# Manganese doping for enhanced magnetic brightening and circular polarization control of dark excitons in paramagnetic layered hybrid metal-halide perovskites

Timo Neumann[1,2], Sascha Feldmann [1], Philipp Moser[2], Alex Delhomme[3], Jonathan Zerhoch[2], Tim van de Goor [1], Shuli Wang[4], Mateusz Dyksik [4,5], Thomas Winkler [1], Jonathan J. Finley [2], Paulina Plochocka [4,5], Martin S. Brandt[2], Clément Faugeras [3], Andreas V. Stier[2] & Felix Deschler [2✉]

Materials combining semiconductor functionalities with spin control are desired for the advancement of quantum technologies. Here, we study the magneto-optical properties of novel paramagnetic Ruddlesden-Popper hybrid perovskites $Mn:(PEA)_2PbI_4$ (PEA = phenethylammonium) and report magnetically brightened excitonic luminescence with strong circular polarization from the interaction with isolated $Mn^{2+}$ ions. Using a combination of superconducting quantum interference device (SQUID) magnetometry, magneto-absorption and transient optical spectroscopy, we find that a dark exciton population is brightened by state mixing with the bright excitons in the presence of a magnetic field. Unexpectedly, the circular polarization of the dark exciton luminescence follows the Brillouin-shaped magnetization with a saturation polarization of 13% at 4 K and 6 T. From high-field transient magneto-luminescence we attribute our observations to spin-dependent exciton dynamics at early times after excitation, with first indications for a Mn-mediated spin-flip process. Our findings demonstrate manganese doping as a powerful approach to control excitonic spin physics in Ruddlesden-Popper perovskites, which will stimulate research on this highly tuneable material platform with promise for tailored interactions between magnetic moments and excitonic states.

[1] Cavendish Laboratory, University of Cambridge, Cambridge, UK. [2] Walter Schottky Institut and Physik Department, Technische Universität München, Garching, Germany. [3] Université Grenoble Alpes, INSA Toulouse, Univ. Toulouse Paul Sabatier, EMFL, CNRS, LNCMI, Grenoble, France. [4] Laboratoire National des Champs Magnétiques Intenses, UPR 3228, CNRS-UGA-UPS-INSA, Grenoble and Toulouse, France. [5] Department of Experimental Physics, Faculty of Fundamental Problems of Technology, Wroclaw University of Science and Technology, Wroclaw, Poland. ✉email: Felix.Deschler@wsi.tum.de

The development of materials which are simultaneously magnetic and semiconducting, while retaining excellent optoelectronic properties and high luminescence yields, is a scientific challenge, which holds great potential for creating novel opto-spintronic functionality for information and communication technologies[1–3]. Dilute magnetic semiconductors (DMS) constitute a material class that combines these properties by introducing magnetic impurities to an otherwise nonmagnetic host semiconductor[4,5]. Inorganic DMS, most prominently transition metal-doped III-V semiconductors, have been known for decades and have enabled new mechanisms, such as highly circularly polarized photoluminescence (PL)[6], spin injection[7], giant magnetoresistance[8] and switching of magnetism by electric fields and currents[9,10]. However, the demanding, mostly epitaxial, material processing techniques as well as limiting material tunability diminish applicability beyond fundamental research[11–13]. Hybrid metal-halide perovskites offer an opportunity for novel control of spin in a high-performance semiconductor due to their exceptional tolerance to structural defects and impurities[14,15], combined with their production as polycrystalline thin films and nanostructures from simple scalable solution-processing techniques[16]. They have created disruptive changes in the field of solution-processed semiconductors for optoelectronics, where they produce highly efficient energy conversion applications, such as solar cells and LEDs[17–19]. High luminescence yields are an exceptional property of the perovskites[20], which are preserved under changes in chemical composition and structure, for example by halide-mixing for tuning of the optical band gap over the visible range[21]. We now show that they hold promise for exciting opportunities in the field of opto-spintronics[22,23]. So far, transition metal-doping has proven useful for altering the optoelectronic properties of lead-halide perovskites, e.g., by enhancing luminescence in Mn-[24], Ni-[25] and Cu-[26] doped nanocrystals, and modified growth and solar cell performance in Mn-, Fe-, Co-, Ni-doped thin films[27,28]. Reports have found signatures of photo-switchable ferromagnetism ($T_C = 25$ K) in single crystals of MA(Mn:Pb)I$_3$ (MA = methylammonium)[29], suggesting a coupling between the dopants' $d$-electron spins and optically excited charge carriers.

Here, we report the paramagnetic manganese-doped 2D Ruddlesden–Popper perovskite Mn:(PEA)$_2$PbI$_4$ (PEA = phenethylammonium, from now on Mn:PEPI and PEPI for the doped and undoped material, respectively) and study its magneto-optical properties. The PL from a magnetically brightened emissive state shows circular polarization up to 13%, which we find to be directly proportional to the material's magnetization. Using transient magneto-luminescence, we attribute this effect to spin-dependent exciton dynamics, which we find to occur within our time resolution (~500 ps) on ultrafast timescales. Our findings constitute the first demonstration of magnetization control of exciton spin physics in a transition metal-doped lead-halide perovskite and provide the first step towards future opto-spintronic functionalities of these materials.

## Results

### Fabrication and characterization of magnetic Ruddlesden–Popper hybrid perovskites

We fabricate Mn:PEPI hybrid perovskite films (Fig. 1a) using an established protocol for PEPI[30] with additional MnX (X = I, Br) precursor (1% atomic ratio Mn relative to Pb). X-ray diffraction measurements and fits of the data to the PEPI crystal structure[31] reveal that Mn-doped and undoped samples are of high crystallinity, showing the typical diffraction pattern of a highly oriented (PEA)$_2$PbI$_4$ film, where the layers of the crystal domains are aligned parallel to the substrate (Fig. S1a). Room temperature absorption and PL spectra show a sharp excitonic resonance at 2.394 eV and a narrow emission peak at 2.353 eV, respectively (Fig. S1b). The absorption and emission of the Mn-doped sample are blue-shifted by 10 and 5 meV, respectively. This is caused by a small bromide content, since MnBr$_2$ was used as the Mn-precursor for solubility reasons, initially. We confirmed the generality of our results presented below also on samples with MnI$_2$ precursor. Low temperature circularly polarized transmission (Fig. 1b) shows evenly spaced minima at 2.356, 2.392, and 2.429 eV which exhibit the same shift with the magnetic field ($\Delta E = \pm 1/2\ g\mu_B B + c_0 B^2$ with $g = 1.1$ and $c_0 = 0.338$ µeV/T$^2$, Fig. S2a) and are attributed to a bright exciton absorption and corresponding phonon replicas[32,33]. We observe neither additional manganese nor magnetic field-induced optically active transitions.

To investigate the magnetic properties of the samples, we carried out electron paramagnetic resonance (EPR) measurements at 4 K (Fig. 1c). We detect a weak response for the pristine sample, likely due to small amounts (ppm) of paramagnetic impurities, showing that the fabricated undoped hybrid perovskites are diamagnetic (Fig. S2b). Upon introducing Mn to the material, a strong and broad resonance appears around a field of 330 mT, corresponding to a $g$-factor of ~2. On top of the single broad resonance, a clear Mn-sextet is observed, which occurs due to the hyperfine coupling of the manganese's $S = 5/2$ electron spin to its nuclear spin of $I = 5/2$. The distinct character of the hyperfine resonances indicates the existence of magnetically isolated Mn$^{2+}$ in the sample, since strong broadening due to interactions occurs in the case of Mn ions in close proximity[34–36]. To confirm the magnetic nature of Mn:PEPI perovskite, the magnetization of the sample was measured with varying temperature and magnetic field using SQUID-magnetometry. For applied fields of 0.01, 0.1, and 1 T the sample shows a $1/T$ Curie-law temperature dependence with no difference between heating and cooling, as is expected for paramagnetic materials (Fig. S2d)[37]. The measured magnetization while sweeping the magnetic field does not show hysteretic behavior and follows the Brillouin function for non-interacting $J = 5/2$ spin systems, corresponding to the spin alignment of the high-spin Mn $d^5$ configuration in the magnetic field (Fig. 1d).

We conclude that we successfully induced paramagnetism to the diamagnetic PEPI perovskite by adding small amounts of manganese salt to the precursor solution. In the following, we investigate the effect of local magnetization of the Mn-moments under external magnetic fields on the exciton recombination in this novel material.

### Magnetic brightening of dark excitons

We employ temperature-dependent PL spectroscopy to investigate the emission spectrum of PEPI and Mn:PEPI at low temperatures. Upon cooling from room temperature, the excitonic peak undergoes a redshift, while two additional peaks emerge below 100 and 10 K (Fig. S1c). At 4 K, the emission spectrum shows three features at 2.342, 2.334, and 2.308 eV, respectively (see Fig. S3a for detailed fitting of the peaks). While the high-energy peak has been well-established as the bright free exciton emission, the lowest energy peak has been proposed to originate from self-trapped excitons, phonon replicas, biexcitons, exciton-polarons or an out-of-plane oriented magnetic dipole transition[38–41]. Since the lowest energy peak is of no importance for the findings of this report, we will not discuss it further.

Importantly, the middle peak at 2.334 eV was identified as a dark state constituting a major loss channel for radiative efficiency[42]. The corresponding peak in PEA$_2$PbBr$_4$ was reported as dark triplet exciton which gains weak oscillator strength by

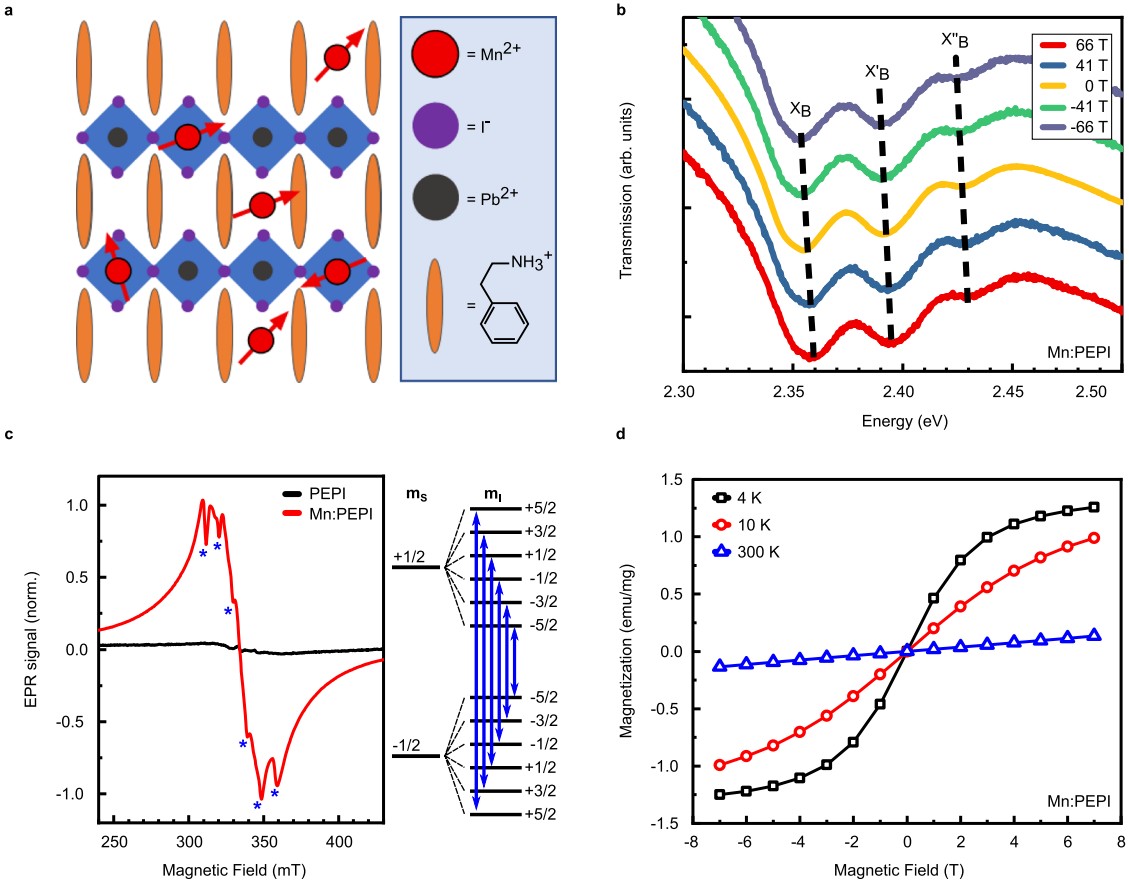

**Fig. 1 Optical and magnetic characterization of Mn:PEPI. a** Sketch of the Ruddlesden–Popper quantum well structure with two possible Mn sites: substitutional for the divalent anion and interstitial in the organic barrier. **b** Circularly polarized optical transmission at 4 K with the magnetic field. The bright exciton and corresponding phonon replica are denoted $X_B$, $X_B'$, $X_B''$, respectively. **c** Electron paramagnetic resonance response of drop casted material at 4 K. Stars denote the six hyperfine transitions. **d** SQUID magnetization versus field sweep at different temperatures. The data shows that $Mn^{2+}$ ions are present in the crystal structure and give rise to paramagnetism without inducing additional optically active trap states.

mixing with a bright state via spin–orbit coupling. In the following, we will refer to the two relevant emission peaks for our magneto-optic studies, at 2.342 and 2.334 eV, as the bright exciton $X_B$ and the dark exciton $X_D$, respectively. Recent studies on exfoliated PEPI single crystals reported that both bright and dark exciton are each split into two substates with ~1 meV energy difference by strong electron–hole exchange interaction, thus yielding an exciton fine structure of four optically-active states[43]. However, since this splitting is likely even smaller in our less confined bulk material and due to larger PL linewidth, we are not able to resolve this fine structure.

We find that, when a magnetic field is applied orthogonal to the quantum well plane (i.e., Faraday configuration), circularly polarized PL spectra ($B > 0$ corresponding to $\sigma^+$ polarization) from pristine and doped perovskite exhibit strong field-dependent effects in intensity and polarization, with different field-response of the two peaks (Fig. 2). The bright exciton peak intensity decreases with the magnetic field (Fig. 2a, b). We note that a fine structure becomes apparent as two new peaks appear approximately 1 meV blue- and red-shifted from the zero-field peak (i.e., Fig. 2a, 14 T), but due to the broad linewidth of the strongly overlapping peaks, this effect cannot be analyzed in greater detail in our study.

Notably, the dark exciton emission greatly increases with the magnetic field to an approximately four-fold enhanced intensity at 14 T, compared to zero-field. While the PL intensity of the dark

exciton peak is greatly enhanced with the magnetic field, the intensity of the bright exciton peak decreases (Fig. 2a inset).

Magnetic brightening of dark states is a known effect in magneto-spectroscopy[44–47]. Commonly it originates from two different mechanisms which are either the shrinking of the excitonic wave function and enhancement of the oscillator strength, or the mixing of spin states allowing the dark state to "borrow" oscillator strength from a bright state via magnetic coupling[48]. Thus, we employ circularly polarized transient photoluminescence spectroscopy under a magnetic field to investigate the detailed nature of the magnetic brightening. The bright exciton transient PL exhibits a short lifetime component which is on the order of our instrument response function (~0.7 ns) and cannot be resolved, while the long component is on the order of ~1 ns, in agreement with literature values[49]. The bright exciton PL kinetics show no detectable change by the application of a magnetic field (Fig. S5a). In contrast, the dark exciton emission shows a strong magnetic field dependence. We find the dynamics at zero field well-described by a biexponential decay with a dominant component with time constant $\tau_{short}$ ~1 ns (close to $\tau_{IRF}$ ~0.7 ns) and a low-weight component with $\tau_{long}$ ~14 ns, characteristic for dark exciton emission. We separately integrate the two exponential decays over time and find that the intensity of the fast component remains constant with the magnetic field, while the intensity of the slow component increases approximately sixfold (Fig. S5c, d). As this increase agrees well with the

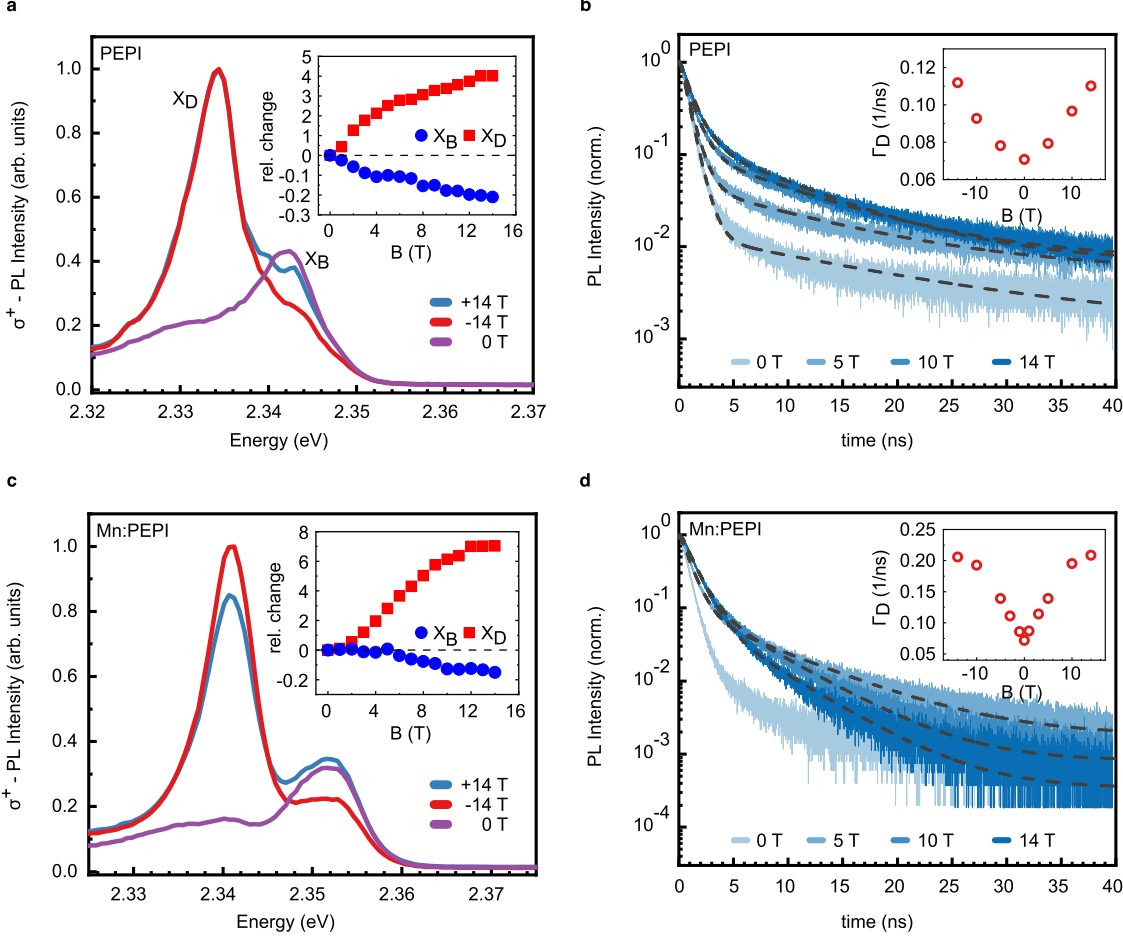

**Fig. 2 Low-temperature magneto-photoluminescence spectra of PEPI and Mn:PEPI. a,c** circularly polarized emission spectra at 4 K of the bright and dark exciton ($X_B$ and $X_D$, respectively) with continuous-wave excitation at 395 nm. Measurements were performed in Faraday geometry with $B = 14$ T, corresponding to $\sigma^+$ detection. Insets: relative intensity change for bright and dark exciton. **b,d** normalized circularly polarized transient PL kinetics of the dark exciton from 0 T to 14 T, for PEPI and Mn:PEPI, respectively. Dashed lines are bi-exponential fits (see Fig. S5 for details of the fitting procedure). Insets: PL emission rate from the long lifetime component of the fit.

observed intensity increase in steady state PL measurements, we confirm the slow decay as the dark exciton emission. Under the applied magnetic field, the PL emission rate increases from 0.07 to 0.11 $ns^{-1}$ with approximately quadratic dependence on field strength (Fig. 2b inset), while the weight of the decay increases ~tenfold (Fig. S5d inset). Due to the higher weight, the slow component dominates the signal at high fields, so that the total PL dynamics decay slower although the slow emission rate increases. The same observations were made in experiments on quantum confined $CsPbBr_3$ to reveal dark exciton emission[50], CdSe nanoplatelets[51] and on single $FAPbBr_3$ nanocrystals, where magnetic brightening was employed to brighten the dark singlet state and reveal the energetic order of exciton states as a dark singlet ground state below a bright triplet[52]. Following the same reasoning, we conclude that the dark exciton in PEPI is brightened by enhanced magnetic coupling to the bright exciton in the presence of strong magnetic fields.

The magneto-PL of the doped Mn:PEPI perovskite (Fig. 2c, d) also shows enhanced magnetic brightening of the dark exciton, but with a higher, seven-fold increase of the dark exciton intensity, and a stronger emission rate increase, which saturates at approximately three times the rate without field (Fig. 2c inset, 2d inset). In contrast to the undoped materials, the emission enhancement does not follow a simple quadratic $B$-dependence. Thus, the manganese-enhanced magnetic brightening is caused

by an additional fast process within our time resolution, e.g., a bright-dark state relaxation, which was proposed for Mn-doped $CsPbCl_3$[53], or by the formation of an additional radiative recombination channel including a spin flip of the Mn moments[54].

The strong absolute increase of PL emission with the magnetic field (integrated intensity +40% (PEPI) / +60% (Mn:PEPI) at 14 T, Fig.3b) shows that the dark state is strongly populated and constitutes a major loss channel in the absence of magnetic-field-induced brightening. This strongly limits the PL quantum efficiency at low temperature, potentially also in similar two-dimensional perovskite systems.

*Polarization control of exciton emission via manganese doping.* Additional to the enhanced brightening, we find that the polarization of the dark exciton PL is greatly affected by the presence of the magnetic dopants. A clear difference in intensity between the positive and negative magnetic field direction is observed, corresponding to circular polarization of the PL (CPL) (Fig. 2c). The bright exciton shows an asymmetry between the field direction (Fig. 3a top), indicating a circularly polarized emission (degree of polarisation DCPL = $[I(\sigma^+) - I(\sigma^-)]/[I(\sigma^+) + I(\sigma^-)]$) with a near-linear dependence (~1.4% / $T$, Fig. S4b) on the applied magnetic field. This magnetic field effect on CPL has been reported in $MAPbI_3$ with a similar slope, and was explained by

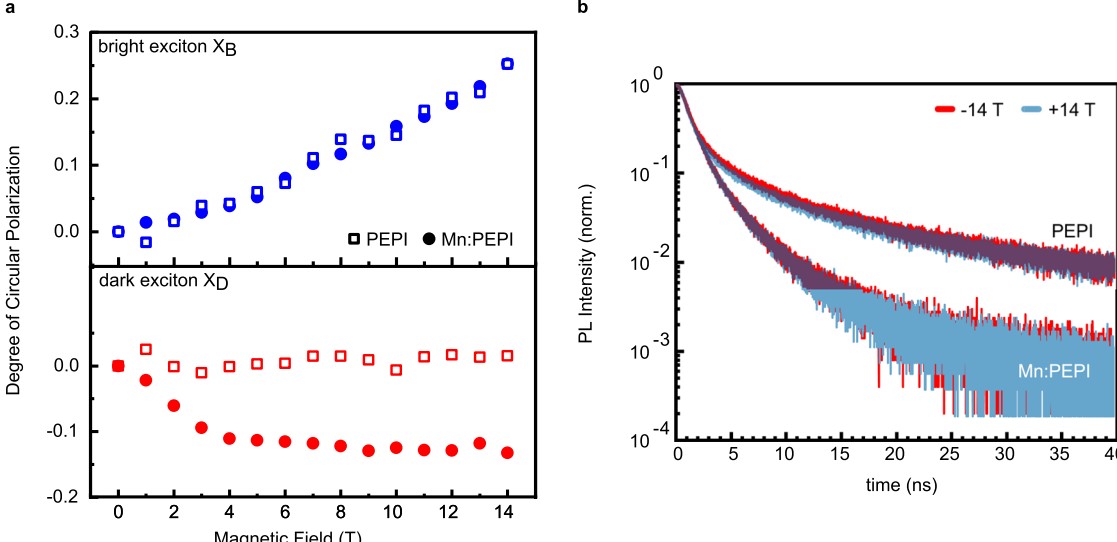

**Fig. 3 Circular PL polarization of PEPI and Mn:PEPI. a** Comparison of undoped PEPI and magnetically-doped Mn:PEPI for the bright (top) and dark (bottom) exciton emission. **b** PL kinetics for opposite field directions, probing the dynamics of circularly polarized emission.

magnetic-field-induced spin-mixing[55] or an imbalanced population of the exciton spin states due to the Zeeman splitting observed in transmission (Fig. 1b).

In the Mn-doped films, the steady state CPL of the dark exciton increases nearly linearly as a function of magnetic field up to ~3 T, then saturates to a constant value (Fig. 3a bottom), while the dark exciton emission of the undoped samples shows no detectable CPL. Notably, the polarization of the bright exciton shows no changes when the magnetic dopant is introduced (Fig. 3a top). Comparison of the absolute slope in the linear region for the dark and bright exciton against the undoped material (~3.0% / $T$ vs. ~1.4% / $T$, Fig. S4b) yields a twofold increase and shows that manganese-induced circular polarization can be tuned more efficiently than the intrinsic field-induced polarization. The observed effect is independent of the excitation polarization and antisymmetric with respect to the magnetic field direction (Fig. S6). We note that the degree of CPL increases with increasing Mn content from 0.5–1%, but shows a local variation (Fig. S7), which we attribute to sample inhomogeneities, i.e., Mn density fluctuation in the sample or local differences in crystal axis orientation. For 2% Mn content, we observe a strongly polarized dark exciton emission, but with a linear magnetic field dependence, supposedly since Mn–Mn interactions become important and couple neighbouring atoms antiferromagnetically[56]. Further investigations on the local crystal structure near Mn-dopants and local PL mapping under magnetic field will be required to resolve details of these effects.

The observed steady state CPL is not reflected in the PL lifetimes of the dark exciton, with PL kinetics showing perfectly overlapping signals for opposite magnetic field directions (Fig. 3b). We, therefore, conclude that the dark exciton CPL originates from a manganese-mediated, exciton spin-dependent transfer or recombination process at timescales well below the resolution of our setup (~500 ps). To get a comprehensive picture of this process, ultrafast magneto-transient absorption and photoluminescence spectroscopy will be required, which is beyond the scope of this work.

Superimposing the optical polarization curve with the magnetization of the sample and a Brillouin function reveals a direct proportionality between the alignment of the magnetic $Mn^{2+}$ dopants' magnetic moment and the polarization of the PL of the paramagnetic perovskite (Fig. 4a). This proportionality strongly suggests coupling of manganese magnetic moments, aligned under external magnetic fields, with the host semiconductor's excitonic states, and confirms that magnetic doping provides a potential approach to control exciton spin state physics in perovskites.

## Discussion

Magnetic transition metal-doping is a well-known approach to induce magnetic properties in inorganic semiconductors. In such dilute magnetic semiconductors, the most significant mechanism is the giant Zeeman effect due to the direct exchange between d-electrons of the localized dopant and the conduction/valence band of the host semiconductor (sp-d exchange)[13,57,58]. The Zeeman energy shift between the different spin-sub-bands is proportional to the magnetization of the material and can be as large as several tens of meV[59,60], which leads to circularly polarized emission from the unequal thermal occupation of the energetically-shifted spin states. Although the polarization of the dark exciton emission in Mn:PEPI follows the magnetization of the material, no giant Zeeman splitting of the PL peaks is observed for the doped and undoped sample. Only a small Zeeman shift is resolved under high magnetic fields (>20 T, Fig. S4a), confirming the existence of spin substates for the dark exciton and suggesting the possibility of a non-zero angular momentum of this state, also in agreement with previous reports on the exciton fine structure in PEPI[43]. We conclude that the sp–d exchange is negligible in Mn:PEPI and does not explain this induced polarization. This observation is in contrast to several transition metal-doped nanostructures, which show polarization of either the excitonic or the Mn d–d transition as well as giant Zeeman splitting caused by the sp–d exchange[6,61,62].

Based on our experimental results, we identify two ultrafast mechanisms as the potential origin of the observed CPL for the dark exciton. In the first scenario (Fig. 4b i), the dark exciton formation rate from the photoexcited excitations changes, as proposed for perovskite nanocrystals[53]. The dark exciton formation might occur from optically excited free carriers or the bright exciton, with the rate $\Gamma_{XF}$ being different for the two dark exciton spin states and proportional to the relative alignment between Mn and exciton spin. Upon recombination, the imbalance in spin state population leads to circular polarized

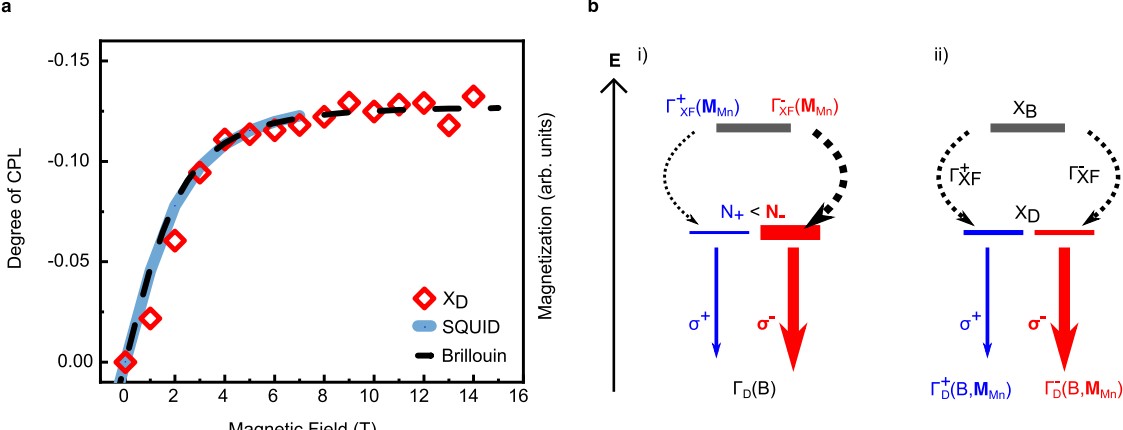

**Fig. 4 Relation between magnetization and CPL polarization in Mn:PEPI and proposed interaction mechanism. a** Superposition of PL polarization and magnetization curve of Mn:PEPI at 4 K and Brillouin function for a non-interacting $J = 5/2$ spin system. **b** Schematic of proposed interaction between excitons and manganese spins in an external magnetic field. $X_B$ and $X_D$ denote bright and dark exciton, respectively. $\Gamma_{XF}$ is the exciton formation rate, N the occupancy of the state, σ the polarization of emission, $\Gamma_D$ is the dark exciton emission rate depending on the magnetic field strength $B$ and the manganese magnetization $\mathbf{M}_{Mn}$. Superscript and subscript $+/-$ denote the corresponding spin state. (i) A magnetization-dependent exciton formation rate favors the occupation of a specific spin state. (ii) Dark exciton spin states are occupied equally, but the manganese induces a spin-dependent recombination pathway leading to different radiative rates.

photoluminescence. In the second scenario (Fig. 4b ii), the manganese modifies the radiative recombination rate of the dark exciton, e.g., by enabling a forbidden transition when the Mn spin flips at the same time when the exciton recombines. This additional radiative pathway has been reported for dangling bond spins as well as for manganese spins[51,54,63,64]. When the manganese spins are aligned in an external magnetic field, this pathway is more efficient for one of the exciton spin states and therefore causes circular polarized photoluminescence. We note that this second scenario explains both, the enhanced emission brightening and the CPL under Mn-doping.

For a better understanding of the underlying mechanism in our materials, a precise knowledge of the Mn lattice site is crucial, but yet to be investigated. The substitution of Pb by Mn was proposed for bulk $CsPbX_3$ (X = Cl, Br)[65], perovskite nanocrystals[24], bulk $MAPbI_3$[29] and $(BA)_2PbBr_4$ (BA = butyl ammonium)[66–68], but interstitial lattice sites, surface decoration or phase segregation, as observed for other dopants, cannot be excluded[65,69,70]. In Ruddlesden–Popper perovskites, the flexibility of the organic layer further allows for intercalation of molecules and ions[71–74]. Knowledge of the Mn site is especially important since the dark exciton might be localized at the organic-inorganic interface[75,76] or at the manganese ion[77], where proximity between dopant and exciton might play a significant role in the origin of CPL. In a comparative study on manganese-doped $CsPb(Cl, Br)_3$ nanocrystals, which exhibit only one excitonic emission peak (Fig. S8), we observe Zeeman splitting as well as circularly polarized emission as a linear function of the magnetic field. However, the magnetic field effects in these nanocrystals are not affected by manganese doping, highlighting the special importance of the nature of excitonic states in our 2D layered structure. Ultrafast spectroscopic techniques under high magnetic fields will be required to reveal the exact spin-dependent photo-physics on the relevant timescales.

To the best of our best knowledge, the reported Mn:PEPI constitutes the first lead-halide perovskite material in which magnetic doping enables direct control of exciton spin physics via a magnetic field, for which we provide evidence for an ultrafast spin-dependent process as the cause of circularly polarized photoluminescence. In contrast to epitaxially fabricated magnetically

doped semiconductors, our hybrid materials allow for introducing magnetic dopants during fabrication from solution processing. Considering the structural and energetic tunability of the perovskite materials, we envision future opportunities for magnetic semiconductors with tailored exciton–dopant interaction and a range of potential applications in spin-based technologies.

## Methods

**Sample fabrication**. All steps of the sample fabrication were carried out in a nitrogen-filled glovebox. The Pb (Mn) precursor solutions were prepared by dissolving $PbI_2$ (LumTech, >99.999%) or $MnBr_2$ (Sigma, >98%) and phenethyl ammonium iodide (LumTech, >99.5%) salts in N,N-dimethylformamide (Sigma, >99.8% anhydrous) with a molar ratio of 1:2. The solutions were stirred for 2 h at 80 °C and filtered with a 0.2 µm pore size PTFE syringe filter. The Pb and Mn precursors were mixed in an atomic ratio of 99:1 to yield the Mn:PEPI solution. Films were prepared on oxygen-plasma-treated glass coverslips by spin coating at 2000 rpm for 30 s and by drop casting and solvent evaporation at 120 °C for 2 h. Samples for the optical measurements were encapsulated.

**EPR**. Electron paramagnetic resonance measurements were carried out on drop casted samples. The setup consisted of a Jeol RE series ESR spectrometer with Jeol JES RE2X power supply and field controller, a Jeol X-Band microwave source, and a $TE_{102}$ (Bruker ER 4102ST) resonator with 100 kHz modulation unit. The temperature in the Oxford ESR 900 cryostat was controlled by a Lake Shore 335 temperature controller. A Stanford Research System Model SR830 DSP lock-in amplifier was used.

**SQUID**. Magnetic properties were measured on a Quantum Design Magnetic Properties Measurement System MPMS XL-7. The sample contained 20 mg of drop casted material in a medical capsule. The sample was cooled from 300 to 2 K with a rate of 10 K/min and the magnetic field was swept −7 to 7 T with a rate of 0.2 T/min. Before each measurement point, the system was given 1 min to equilibrate.

**Magneto-PL**. Magneto photoluminescence measurements were carried out in the Laboratoire National des Champs Magnétiques Intenses-Grenoble (LNCMI-G). The samples were installed in a closed tube filled with helium exchange gas and cooled down to 4 K. The excitation source was a 395 nm (3.139 eV) diode laser guided through a fiber and focused onto a ~5 µm spot with an excitation power of 1 µW (5.09 W/cm$^{-2}$) on the sample. In the detection path, a right-handed polariser (σ$^+$) was mounted. The magnetic field was swept from 14 to −14 T with a step size of 0.25 T. For each magnetic field, three spectra were obtained with an integration time of 10 s per spectrum. Transient photoluminescence has been measured with the same optical fiber-based experimental set-up. The signal was collected using an avalanche photodiode (picoquant) coupled to a 50 mm focal length spectrometer. A pico-second laser emitting at $\lambda = 434$ nm was used for excitation and the temporal resolution was 500 ps.

**XRD.** X-ray diffraction measurements were made on thin-film samples on a Bruker D8 discover diffractometer with Cu Kα radiation, $\lambda = 1.5403$ Å. Samples were measured using a Bragg-Brentano geometry over $5 \le 2\theta \le 65$ with a step size of $\Delta 2\theta = 0.001$. Patterns were fitted using the LeBail method in TOPASv5. The background was modeled with a Chebyshev polynomial function of 3rd order and the peak shape was set to Thompson–Cox–Hasting pseudo-Voigt. In order to account for the strong preferential orientation, only the c lattice parameter and zero offset were refined.

**UV/VIS and PL.** Absorption spectra of spin-coated films were measured using a Shimadzu UV600 spectrometer in linear transmission mode. PL spectra were measured on an Edinburgh Instruments FLS90 fluorimeter.

**Transmission in high magnetic field.** Transmission spectra as a function of the magnetic field were measured in a pulsed field magnet with maximum field B = 66 T and pulse duration of ≈ 500 ms. Broad-band white light was provided by a tungsten halogen lamp. The magnetic field measurements were performed in the Faraday configuration, with the c-axis of the sample parallel to the magnetic field and incident light. The circular polarization was resolved in situ using a quarter waveplate and a polarizer. Measuring in both directions of the magnetic field provides access to the σ + and σ − polarized states. The light is sent to the sample using an optical fiber. The transmitted signal is collected by a lens and coupled to another fiber. The signal is dispersed using the grating of a monochromator and detected using a liquid-nitrogen-cooled CCD camera. The sample is placed in a liquid helium cryostat and cooled down to 2.2 K.

## Data availability statement

The data that support the findings of this study are available in the "mediaTUM" repository under the DOI: "https://doi.org/10.14459/2021mp1609815".

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

## Acknowledgements

T.N. acknowledges funding from the Winton Programm for the Physics of Sustainability. S.F. acknowledges funding from the Studienstiftung des deutschen Volkes and the Engineering and Physical Sciences Research Council via a PhD and Doctoral Prize Fellowship, as well as support from the Winton Programme for the Physics of Sustainability. A.V.S. and P.M. gratefully acknowledge the German Science Foundation (DFG) for financial support via the Cluster of Excellence e-conversion EXS 2089. T.v.d.G. acknowledges support from the EPSRC Cambridge NanoDTC, EP/L015978/1 and funding from the Schiff Scholarship. Part of this work was performed at the LNCMI, a member of the European Magnetic Field Laboratory (EMFL). F.D. acknowledges funding from the Winton Programm for the Physics of Sustainability, the DFG Emmy Noether Program, and an ERC Starting Grant. This project has received funding from the European Research Council (ERC) under the European Union's Horizon 2020 research and innovation program (grant agreement No 852084. P.P. acknowledges the support from National Science Centre Poland within the MAESTRO program (grant number 2020/38/A/ST3/00214). This work was partially supported by OPEP project, which received funding from the ANR-10-LABX-0037-NEXT. M.D. appreciates the support from the Polish National Agency for Academic Exchange within the Bekker program (grant no. PPN/BEK/2019/1/00312/U/00001). The Polish participation in European Magnetic Field Laboratory is supported by the DIR/WK/2018/07 grant from the Ministry of Science and Higher Education, Poland.

## Author contributions

T.N., S.F., and F.D. conceived and planned the experiments. T.N. fabricated the samples, performed X-ray diffraction, SQUID magnetometry, absorption spectroscopy and initial magneto-spectroscopy with input from S.F., T.v.d.G., and T.W. T.v.d.G. fitted the XRD data. J.Z. performed EPR measurements with input from M.S.B. P.M., A.D, A.S., and C.F. performed low-temperature magneto-PL spectroscopy at LNCMI Grenoble. S.W. and M.D. performed the high-magnetic-field measurements at LNCMI Toulouse, supervised by P.P. T.N. and F.D. drafted the manuscript and compiled figures, with the discussion of results and feedback on the manuscript from all authors.

## Funding

## Competing interests

The authors declare no competing interests.
