## [Peer Review File · Nature Communications]

Reviewers' Comments:

Reviewer #1:

Remarks to the Author:

The work titled "Magnetic proximity effect on excitonic spin states in Mn-doped layered hybrid perovskites" by Neumann et al. reports and discuss the effect of Mn doping into 2D perovskites. Overall, the findings presented in this work are novel and interesting. However, I find that this work is poorly presented, in terms of both graphical presentations and scientific analysis. For instance, the PL peaks are not fitted and well resolved for different emitting states; many observations are brushed off and left unexplained; etc. For comparison on the presentation quality, I would refer the Authors to similar work by Do et al., Nano Lett. 2020, 20, 7, 5141–5148.

Scientific-wise, there are several issues that have to be resolved, before I could recommend the publication in Nature Communications:

Major issues:

1. In Figure 3a top, why would the circular PL in the Mn-doped sample initially decrease, before it changes to be increasing around 3 T? In Figure S3, why would one observe opposite circular PL behaviors for free and localized excitons? Both Figure 3 and Figure S3 should show the same experiment. However, the results presented are totally different in these two figures, especially the free exciton trend with B-field for the Mn-doped sample.

2. The mechanism of such circular PL mechanism from the Authors is too qualitative to be accepted. The Authors simply assigned it to the proximity dipole-dipole interaction between exciton and the manganese dopant. However, there is no further measurement or theory that backs this assignment. For instance, why would dipole-dipole interaction (electric coupling) would affect the spin polarization (magnetic coupling)? Or could the Authors verify the assignment of dipole-dipole interaction (proportional to $1/r^3$, where r is the interparticle distance), with the dopant concentration dependent measurement?

3. Another possible assignment would be the Mn dopant induced the formation of a spin-polarized/spin-dependent trap states, which localized the exciton. Could the Authors disprove this assignment?

4. In page 7 Line 137, the Authors assigned the magnetic brightening to either the increase of transfer rate from bright to dark exciton or the increase in the dark exciton recombination rate. However, the Authors leave the investigation of such matter in the future work. In the light of previous comment, I believe time-resolved measurement is relevant and have to be reported in the current work, as it might bear consequence to the overall interpretation of the system.

Minor issues:

5. The definition of circular PL is not defined. Is it $(\sigma^+ - \sigma^-)/(\sigma^+ + \sigma^-)$?

6. Why was the Mn doping on the perovskite performed using MnBr₂ and not MnI₂? While it is less likely to be an issue, the conclusion taken is strictly speaking unjustified. since the effect might come as well from the Bromide effect, instead of due to pure Manganese effect.

7. In the methods and figure 2 captions, the author mentioned that the detection was set to σ^+ circular detection. However it is not clearly mentioned in the manuscript.

Reviewer #2:

Remarks to the Author:

The authors report the increase in circular PL polarization anisotropy in Mn-doped Ruddlesden-Popper hybrid perovskite compared to without Mn doping. The circular PL polarization under external magnetic field, enhanced by the doped Mn, is stronger for the PL from the magnetically brightened 'localized' exciton than free exciton. The authors attribute the stronger effect of paramagnetic dopant on circular PL polarization of the localized exciton to its proximity to Mn. Although this study reports an interesting aspect of magneto-optic properties of Mn-doped layered perovskites, the impact of the findings and the level of understanding of the phenomenon are limited to recommend this work for publication in Nature communications.

1) For this work to be more impactful, the origin of the center PL assigned as 'localized' exciton by the authors should be established more firmly. From the magnetic brightening of this PL, the authors infer dark exciton state. The authors should perform time resolved PL study for clearer identification of the emitting state instead of deferring it for a later study. I am particularly concerned about the uncertainty of the identity of this state, especially when the PL intensity increases dramatically via magnetic brightening, dominating the total PL intensity at the higher magnetic field. Why does magnetic brightening enhance the overall PL intensity so much? How does the overall PL quantum yield (QY) change with magnetic field? Can such large increase of total PL intensity be explained by intensity borrowing from bright state by magnetic mixing or by wavefunction shrinking? Before discussing the control of exciton spins states by paramagnetic dopants, proved via circular polarization of the PL, the identity of the state in question should be more firmly established.

2) The effect of magnetic dopant on the exciton PL appears as an additional circular PL anisotropy, which is 2.5 time larger for 'localized' exciton than free exciton. The authors explain this difference with the closer proximity between localized exciton and Mn than between free exciton and Mn. Attributing the observed difference simply to difference in spatial proximity seems an oversimplification and the hypothesis is not well backed up by experimental or theoretical evidence. The conclusion on the mechanism is too speculative.

3) Why does the 'localized' exciton PL's circular polarization show no response to the external magnetic field in the absence of Mn in contrast to free exciton PL? Why is the sign of circular polarization of localized and free exciton opposite?

4) For free exciton PL, the major component of circular polarization originates from the external field at the higher field intensity ($>10T$), which increases linearly with magnetic field. In contrast, the circular polarization of localized exciton saturates above 6T. What is the implication of this early saturation and the opposite sign of the circular PL polarization between free and localized excitons in overall controllability of the exciton spin in the application in spin-based technology?

5) The authors proposed localization of exciton at the interface and interlayer Mn doping rather than substitutional doped within the layer to explain the observation. It also seems the possibility of substitutional doping is also precluded since it could be incompatible with the argument based on exciton-dopant spatial proximity. The authors will need to provide some evidence to support their hypothesis to avoid being overly speculative in their conclusion.

Reply to Reviewer Comments

Reviewer #1 (Remarks to the Author):

The work titled “Magnetic proximity effect on excitonic spin states in Mn-doped layered hybrid perovskites” by Neumann et al. reports and discuss the effect of Mn doping into 2D perovskites. Overall, the findings presented in this work are novel and interesting. However, I find that this work is poorly presented, in terms of both graphical presentations and scientific analysis. For instance, the PL peaks are not fitted and well resolved for different emitting states; many observations are brushed off and left unexplained; etc. For comparison on the presentation quality, I would refer the Authors to similar work by Do et al., Nano Lett. 2020, 20, 7, 5141–5148.

We thank the reviewer for their positive assessment on novelty and interest of our work. We apologize that the analysis appeared sub-par and we have worked on improving presentation, discussion and added further data to strengthen our findings. We have taken on their constructive comments, which helped us to improve the quality of the manuscript, for which we are grateful.

In detail, we have:

- Discussed the measured peaks based on acquired magneto-absorption data, as well as existing literature.
- Performed fits to the PL spectra and used the fitted peak intensities for a clearer analysis of the magnetic field effects.
- Acquired and included further data for transient magneto-PL, which now provides a first understanding of the origin of our observations (see also detailed response below).
- Improved the presentation and layout of all figures.

Scientific-wise, there are several issues that have to be resolved, before I could recommend the publication in Nature Communications:

Major issues:

1. In Figure 3a top, why would the circular PL in the Mn-doped sample initially decrease, before it changes to be increasing around 3 T? In Figure S3, why would one observe opposite circular PL behaviors for free and localized excitons? Both Figure 3 and Figure S3 should show the same experiment. However, the results presented are totally different in these two figures, especially the free exciton trend with B-field for the Mn-doped sample.

We thank the referee for pointing out this discrepancy and the helpful comment on better fitting procedures of the peaks. We have performed the suggested advanced fitting analysis now, which helps us to distinguish overlaid effects. The old versions showed the CPL over the wavelength range of the free exciton, which we now find to be a superposition of a bright exciton peak X_B , a dark exciton peak X_D , and a weak, lower energy emission band, which has been discussed in literature (Figure S3). In our revised analysis, we can now resolve the individual contributions and find that the initial decrease in circular polarization of the, now called, bright

exciton emission was due to overlaid signals from the dark exciton (Figure 3a). The bright exciton shows purely the expected linear increase expected from the Zeeman splitting we now resolve in the added magneto-absorption data. In contrast the dark exciton shows a behavior that follows the magnetization curve of the material (Figure 3a, bottom). We resolve the detailed origin of this main result from transient luminescence (see reply to reviewer comment #2 and #4 below).

We further note that Fig 3 and S3 show the same experiment, but on a different sample batch. The general trend is the same, while the magnitude of the effect is different, which we attribute to sample inhomogeneities, as discussed in the manuscript, which we expect to be overcome with future optimization of the synthesis.

2. The mechanism of such circular PL mechanism from the Authors is too qualitative to be accepted. The Authors simply assigned it to the proximity dipole-dipole interaction between exciton and the manganese dopant. However, there is no further measurement or theory that backs this assignment. For instance, why would dipole-dipole interaction (electric coupling) would affect the spin polarization (magnetic coupling)? Or could the Authors verify the assignment of dipole-dipole interaction (proportional to $1/r^3$, where r is the interparticle distance), with the dopant concentration dependent measurement?

We agree with the referee that further clarification on the origin of the CPL is required, for which we have now taken transient PL data under high fields (see detailed discussion to comment #4). In short, we explain the CPL for the bright exciton from a Zeeman effect and find that an ultrafast process within the first nanosecond is the origin of the measured CPL of the dark exciton, likely due to spin-selectivity in an initial transfer or recombination processes. Further experiments with ultrafast time-resolution are planned to resolve this mechanism in full detail, but are currently beyond the scope of our capabilities.

We further added concentration dependent measurements and probe the trends in CPL on increasing Mn content (Figure S7). These measurements confirm our result of a Mn-induced origin of the CPL, i.e. an increased degree of CPL with increasing Mn content. The results do not follow the $1/r^3$ dependence proposed by the referee, which indicates a more complex behavior, likely due to increasing inhomogeneity from aggregation of Mn and local differences in crystal alignment at higher doping concentrations. Further investigations on fabrication and local environment of the Mn-dopants are under way (e.g. NEXAFS experiments), but beyond the scope of the current manuscript.

We have now extensively reworded our interpretation of the origin of the CPL based on the suggestions of the referee and our additional transient CPL data, which allows us to go beyond the initial hypothesis on magnetic proximity effects.

3. Another possible assignment would be the Mn dopant induced the formation of a spin-polarized/spin-dependent trap states, which localized the exciton. Could the Authors disprove this assignment?

We expect trapping to be a less relevant process in our samples, since we do not form a new emission peak upon Mn doping, but purely brighten an existing one and change the polarization of this state. Thus, we find that speaking of an Mn induced trap would be misleading. But we agree that, more generally, localization of the dark exciton near Mn sites is a likely process, which we have already discussed in the manuscript. We see it as a likely explanation for stronger interactions between the dark exciton and Mn-dopants than of the free exciton, which would provide a rationalization of the ultrafast processes (see also detailed reply on recombination dynamics on comment #4 below), which we now identify as the source of the measured CPL. We have revised the section of the manuscript to clarify these aspects:

Based on our experimental results, we identify two ultrafast mechanisms as the potential origin of the observed CPL for the dark exciton. In the first scenario (Fig. 4b i), the dark exciton formation rate from the photoexcited excitations changes, as proposed for perovskite nanocrystals.¹ The dark exciton formation might occur from optically excited free carriers or the bright exciton, with the rate Γ_{XF} being different for the two dark exciton spin states and proportional to the relative alignment between Mn and exciton spin. Upon recombination the imbalance in spin state population leads to circular polarized photoluminescence. In the second scenario (Fig. 4b ii), the manganese modifies the radiative recombination rate of the dark exciton, e.g. by enabling a forbidden transition when the Mn spin flips at the same time when the exciton recombines. This additional radiative pathway has been reported for dangling bond spins as well as for manganese spins.²⁻⁵ When the manganese spins are aligned in an external magnetic field, this pathway is more efficient for one of the exciton spin states and therefore causes circular polarized photoluminescence. We note that this second scenario explains both, the enhanced emission brightening and the CPL under Mn-doping.

(Page 13 lines 13 ff.)

4. In page 7 Line 137, the Authors assigned the magnetic brightening to either the increase of transfer rate from bright to dark exciton or the increase in the dark exciton recombination rate. However, the Authors leave the investigation of such matter in the future work. In the light of previous comment, I believe time-resolved measurement is relevant and have to be reported in the current work, as it might bear consequence to the overall interpretation of the system.

We thank the referee for their previous comments and agree with the need for further experiments. We have now performed transient magneto-PL experiments to clarify the process of magnetic brightening. These results have now been added as Figure 2b/d, Figure 3b and Figure S5.

In the undoped films, the bright exciton recombination kinetics are unaffected by an external magnetic field (Figure S5a), while the dark exciton emission rate increases quadratically with magnetic field (Figure 2b). This shows that the origin of the brightening of the emission can be explained by coupling to a bright state via state-mixing, comparable to the case in FAPbBr₃ and CsPbBr₃ nanomaterials,^{6,7} and in agreement with the decrease in overall integrated bright exciton emission (Figure 2 a).

In the Mn-doped films we resolve now a stronger increase in the emission rate for the dark exciton, still without changes in the bright exciton emission kinetics (Figure 2c/d, Figure S5b).

Thus, we resolve the additional brightening of the dark exciton emission as an additional Mn-induced process, which we attribute likely to be a Mn-facilitated spin-flip mechanism.

Further, the transient PL data now provides mechanistic insights into the origin of the CPL origin of the dark exciton (we assign the bright exciton CPL to classical Zeeman splitting, as discussed above). We find no difference in lifetime for the two polarizations of the dark exciton emission, suggesting an origin of the anisotropy as a spin-selective conversion/transfer process from initially excited states to the dark exciton within the time-resolution of our PL experiment. Resolving the detailed processes involved in the transfer will require complex experiments with higher temporal resolution, e.g. magneto-pump-probe experiments, which we are planning in the future, but which are currently beyond our capabilities.

Minor issues:

5. The definition of circular PL is not defined. Is it $(\sigma^+ - \sigma^-)/(\sigma^+ + \sigma^-)$?

We have added this now to the manuscript on page 11 line 5.

degree of polarization DCPL = $[I(\sigma^+) - I(\sigma^-)]/[I(\sigma^+) + I(\sigma^-)]$

6. Why was the Mn doping on the perovskite performed using MnBr₂ and not MnI₂? While it is less likely to be an issue, the conclusion taken is strictly speaking unjustified. since the effect might come as well from the Bromide effect, instead of due to pure Manganese effect.

This was due to an initial solubility issue for MnI₂, which was resolved during the review process. We added now measurements on samples from MnI₂ precursor, which confirm that our results do not depend on the precursor (Figure S7a). However, we decided to keep the results from samples with MnBr₂ precursor in the paper since we found that with this fabrication the reported effects are most prominent. This is likely due to better solubility of MnBr₂, which allows for more facile incorporation of Mn.

7. In the methods and figure 2 captions, the author mentioned that the detection was set to σ^+ circular detection. However, it is not clearly mentioned in the manuscript.

We have added this now to the manuscript on page 7 line 18.

Reviewer #2 (Remarks to the Author):

The authors report the increase in circular PL polarization anisotropy in Mn-doped Ruddlesden-Popper hybrid perovskite compared to without Mn doping. The circular PL polarization under external magnetic field, enhanced by the doped Mn, is stronger for the PL from the magnetically brightened 'localized' exciton than free exciton. The authors attribute the stronger effect of paramagnetic dopant on circular PL polarization of the localized exciton to its proximity to Mn. Although this study reports an interesting aspect of magneto-optic properties of Mn-doped layered perovskites, the impact of the findings and the level of understanding of the

phenomenon are limited to recommend this work for publication in Nature communications.

We thank the referee for assessing that our study resolves an 'interesting aspect of magneto-optic properties of Mn-doped layered perovskites'. We take on the constructive criticism on the level of understanding provided in the initial submission, which we have now substantially expanded with further magneto absorption and transient magneto-luminescence data. With this further analysis, we can now provide a better level of understanding of the mechanistic origin of the observed magnetic brightening and the polarized emission. As requested by the referee, we are confident that these additions substantially strengthen the impact of our study. Please see our responses to the comments below for full details.

1) For this work to be more impactful, the origin of the center PL assigned as 'localized' exciton by the authors should be established more firmly. From the magnetic brightening of this PL, the authors infer dark exciton state. The authors should perform time resolved PL study for clearer identification of the emitting state instead of deferring it for a later study. I am particularly concerned about the uncertainty of the identity of this state, especially when the PL intensity increases dramatically via magnetic brightening, dominating the total PL intensity at the higher magnetic field.

We thank the referee for this constructive comment and the suggestion to perform transient CPL experiments to add impact to our study. We have now performed transient magneto-PL experiments to clarify the process of magnetic brightening. These results have now been added as Figure 2b/d, Figure 3b and Figure S5. We have further performed fitting of the PL peaks, as also requested by reviewer 1, and find two peaks, which we now label bright and dark exciton (based on their behavior under applied magnetic field in transmission and transient PL).

In the undoped films, the bright exciton recombination kinetics are unaffected by an external magnetic field (Figure S5a), while the dark exciton emission rate increases quadratically with magnetic field (Figure 2b). This shows that the origin of the brightening of the emission can be explained by a bright state via state-mixing, comparable to the case in FAPbBr_3 and CsPbBr_3 nanomaterials,^{6,7} and in agreement with the decrease in overall integrated bright exciton emission (Figure 2 a).

In the Mn-doped films we resolve now a stronger increase in the emission rate for the dark exciton, still without changes in the bright exciton emission kinetics (Figure 2c/d, Figure S5b). Thus, we resolve the additional brightening of the dark exciton emission as an additional Mn-induced process, which we attribute likely to be a Mn-facilitated spin-flip mechanism.

We have now added magneto-transmission data which provides a more detailed picture of the nature of the states.

From the observed Zeeman shift of the bright exciton, we extract a g-factor and diamagnetic shift constant of $g = 1.1$ and $c_0 = 0.338 \mu\text{eV}/\text{T}^2$. The dark exciton is not observed in absorption, even with applied magnetic field. This shows that the dark exciton is indeed a forbidden transition to the ground state, while the bright exciton is a manifold of spin-orbit coupled states with bright transitions to the ground state. We further added more citations to recent papers in our manuscript, which have focused on the investigation of the nature of the exciton states and provide further discussion of the assignment of the observed peaks.

With this additional data and the related analysis, we now present a more complete picture of the mechanism of the Mn-induced emission brightening and the CPL (see reply to comment #2 below), which are at the core of our report. With these modifications and additions, as recommended by the referee, we are confident that we increased the impact of our findings to be on the level required for Nature Communications.

Why does magnetic brightening enhance the overall PL intensity so much? How does the overall PL quantum yield (QY) change with magnetic field? Can such large increase of total PL intensity be explained by intensity borrowing from bright state by magnetic mixing or by wavefunction shrinking?

Based on the transient magneto-PL data, we can distinguish two different origins for the increased PL intensity. In the undoped materials the PL lifetime of the dark exciton is enhanced under magnetic field, indicating intensity borrowing from the bright exciton by magnetic mixing. This is confirmed by the expected quadratic field dependence of emission rate (Figure 2b).

For the doped samples we find a stronger increase of the emission rate and a deviation from the quadratic field dependence. Combined with the concomitant increase in overall PL, this indicates that the interaction with the Mn-dopants leads to an increase in the radiative rates, likely related to changes in the wavefunction from exciton localization and a potential Mn-mediated spin-flip, as suggested by the referee, and in agreement with our discussions in the initial submission. We have expanded the related discussion now in the manuscript on page 9.

Before discussing the control of exciton spins states by paramagnetic dopants, proved via circular polarization of the PL, the identity of the state in question should be more firmly established.

Based on the added magneto-transmission, transient magneto-luminescence and further literature review (see our detailed reply above), we have now identified the relevant aspects of the states in question to draw up a mechanistic picture of the photophysical processes leading to brightening and CPL. While further details on the, e.g. structural or very detailed electronic, nature of the states are of fundamental interest, these are not required for our ability to present a mechanistic origin of our observed effects now.

2) The effect of magnetic dopant on the exciton PL appears as an additional circular PL anisotropy, which is 2.5 time larger for 'localized' exciton than free exciton. The authors explain this difference with the closer proximity between localized exciton and Mn than between free exciton and Mn. Attributing the observed difference simply to difference in spatial proximity seems an oversimplification and the hypothesis is not well backed up by experimental or theoretical evidence. The conclusion on the mechanism is too speculative.

We have now resolved the mechanistic origin for the CPL in more detail from the magneto-absorption and transient PL experiments.

The transient PL data provides mechanistic insights into the origin of the CPL origin of the dark exciton (The bright exciton shows purely the expected linear increase in CPL expected from the Zeeman splitting we now resolve in the added magneto-absorption data). We find no difference in lifetime for the two polarizations of the dark exciton emission, suggesting an origin of the anisotropy as a spin-selective conversion/transfer process from initially excited states to the

dark exciton within the time-resolution of our PL experiment. Resolving the detailed processes involved in the transfer will require complex experiments with higher temporal resolution, e.g. magneto-pump-probe experiments, which we are planning in the future, but which are currently beyond our capabilities.

We further added concentration dependent measurements and probe the trends in CPL on increasing Mn content (Figure S7). These measurements confirm our result of a Mn-induced origin of the CPL, i.e. an increased degree of CPL with increasing Mn content. The results do not follow the $1/r^3$ dependence proposed by the reviewer 1, which indicates a more complex behavior, likely due to increasing inhomogeneity from aggregation of Mn and local differences in crystal alignment at higher doping concentrations. Further investigations on fabrication and local environment of the Mn-dopants are under way (e.g. NEXAFS experiments), but beyond the scope of the current manuscript.

3) Why does the 'localized' exciton PL's circular polarization show no response to the external magnetic field in the absence of Mn in contrast to free exciton PL? Why is the sign of circular polarization of localized and free exciton opposite?

To clarify this point, we performed magneto transmission experiments. We find that the bright exciton shows a Zeeman splitting (Figure 1b) and associate its CPL (in undoped and doped samples) with imbalanced equilibrium occupancy due to this energetical difference. The dark exciton does not appear in transmission, but perfectly overlapping PL peaks for positive and negative fields suggest that no detectable Zeeman splitting occurs. Thus, spin states are equally occupied and no CPL occurs.

From transient PL experiments we can further rule out a difference in the slow, magnetic field enhanced recombination rate as the origin of the CPL of the dark exciton (Figure 3b). Thus, we conclude that the dark exciton CPL is caused by an ultrafast spin dependent transfer or recombination process within our time resolution of ~ 500 ps, rather than energetic difference between exciton spin states. Since the effect only occurs under Mn-doping we interpret our results as a Mn-mediated spin-flip process at very early times. Full clarification of the mechanisms on these ultrafast timescales are planned from pump-probe experiments at high fields, but currently beyond the scope of this initial report, which focuses on the unexpected observation that Mn-doping can control the exciton photo-physics in hybrid perovskites. For this we now provide a clear assignment of the involved excitonic states and the underlying recombination dynamics, from which we can propose a general mechanism of the observed effects.

Since we can now identify that bright and dark exciton CPL originate from fundamentally different processes, there is no reason why both effects necessarily have the same sign.

4) For free exciton PL, the major component of circular polarization originates from the external field at the higher field intensity (>10 T), which increases linearly with magnetic field. In contrast, the circular polarization of localized exciton saturates above 6T. What is the implication of this early saturation and the opposite sign of the circular PL polarization between free and localized excitons in overall controllability of the exciton spin in the application in spin-based technology?

We now provide a clear mechanistic origin of these observations, i.e. bright exciton CPL is caused by a Zeeman splitting, while dark exciton CPL originates from a Mn-induced spin-flip process at early times.

In terms of applications, spin-based devices require strong spin polarization at low magnetic fields, for which we find the strongest dependence of our observed effects. We now also compare the relative dependence of the CPL on magnetic field in the manuscript and find that the dark exciton shows a steeper increase than the bright exciton CPL at low fields.

Further, the opposite sign of the CPL is of no importance, since the two excitons can easily be spectrally separated due to the large energetic offset between the transitions.

5) The authors proposed localization of exciton at the interface and interlayer Mn doping rather than substitutional doped within the layer to explain the observation. It also seems the possibility of substitutional doping is also precluded since it could be incompatible with the argument based on exciton-dopant spatial proximity. The authors will need to provide some evidence to support their hypothesis to avoid being overly speculative in their conclusion.

We now focus our discussion stronger on the additional insights from transient PL, which shows that the cause of the dark exciton CPL must be an ultrafast transfer or recombination process in the presence of Mn-dopants. We attribute the origin of this effect to an Mn-induced spin flip process that acts particularly on the dark exciton, due to its different spin nature compared to the bright exciton, as evidenced by our additional magneto-absorption and PL experiments. We have further added data on the CPL dependence on Mn-concentration (Figure S7), which shows an increase with concentration, confirming our proposed model.

We now discuss the possibilities of both, substitutional and interlayer doping, as the origin of the dark exciton interaction, since our mechanism does not require a special structural property of the dark exciton anymore. We have reworded this section of the manuscript and now discuss the origin of the dark exciton CPL from a Mn-induced spin flip process, without discussing a detailed structural origin.

While not relevant for our current report, we note that, additional experiments are under way to determine the Mn-doping site, for example via NEXAFS. The demonstration of Mn-dopant control of excitonic emission in the layered hybrid perovskites is an exciting finding, for which we now provide ample evidence of the underlying mechanisms and involved excitonic species, which provides the required insight for the wider field to start exploiting the broad possibilities for fundamental studies and applications based on this class of hybrid materials.

Reviewers' Comments:

Reviewer #1:

Remarks to the Author:

I appreciate the Authors for making a significant improvement on the manuscript, especially in terms of the analysis of the CPL mechanisms. The Authors have also answered most of my previous issues on the manuscript. There are several other questions I want to post to the Reviewer regarding their time-resolved studies:

1. The Authors claimed observation of the increased PL emission rate of the dark exciton (XD) with B-field (line 165). However, what I saw in Fig. 2b is the reverse, where the PL dynamics without B-field (light blue) decay much faster than with B-field (dark blue), i.e., thus a decreasing rate with B-field. This contradiction between the observation and interpretation cast doubt on the whole manuscript.

2. Moreover, If the Authors use a bi-exponential fitting in explaining the dynamics, I suggest the Authors either report each lifetime component and give interpretation to them or report the rate as the effective lifetime. However, this should be clearly stated in the manuscript together with the rationale of why the Authors do so. The Authors should also show that such analysis would not affect the further interpretations down the line.

3. Given the comments #1 and #2 can be answered, I like the Authors' new interpretation of two possible mechanisms. Could I suggest the Authors also do quantitative modelling on their interpretations to fit their experimental data and provide the quantitative values for the spin-dependent XD formation rate and/or XD recombination rate, based on the 2 proposed mechanisms?

Unless the Authors could provide a satisfactory answer on comments #1 and #2, I could not recommend this paper for publication. On a side note, if the case if this work could be published in the future, I believe the data presentation could be further improved (e.g., Fig. 3a is too cluttered). Moreover, I believe some figures in the SI should also go to the main text (e.g., Fig. S3 and Fig. S6a-b).

Reviewer #2:

Remarks to the Author:

I appreciate the authors for performing additional experiments and adding more data. The additional data certainly helped identifying the origin of the emission peaks in the spectra. Although many points from the earlier review are clarified, I am concerned about the possible mechanisms of CPL of dark exciton the authors presented in this revision.

The PL assigned to dark exciton in PEPI (Fig. 3a) shows no CPL in contrast to bright exciton, which seems consistent with singlet dark state ($J=0$) and triplet bright state ($J=1$) reported by Efros and coworkers (Nature volume 553, pages189–193(2018)) for cesium lead halide perovskites. In contrast, the possible mechanisms for CPL in Mn:PEPI the authors presented, and in Fig.4, dark exciton with different spin states are mentioned implying nonzero J for dark exciton in Mn:PEPI. Although the detailed spin-dependent physics may be done in future, the feasibility of dark exciton to have different spin states should be discussed, because it is crucial for understanding the mechanism of the polarization control which is the central theme of this study. Of course, 2D layered perovskite may have different exciton fine structure from 3D nanocrystals studied in the work by Efros and coworkers. If this structural difference results in nonzero J for dark exciton, it should be discussed, although the data from PEPI seems to suggest J is still zero in 2D perovskite.

Reply to Reviewer Comments

Reviewer #1 (Remarks to the Author):

I appreciate the Authors for making a significant improvement on the manuscript, especially in terms of the analysis of the CPL mechanisms. The Authors have also answered most of my previous issues on the manuscript. There are several other questions I want to post to the Reviewer regarding their time-resolved studies:

We thank the reviewer for attesting significant improvement of our manuscript from our additional experiments, analysis and discussion. Following their recent comments, we are confident to have now addressed all remaining open questions, as outlined in the following.

1. The Authors claimed observation of the increased PL emission rate of the dark exciton (XD) with B-field (line 165). However, what I saw in Fig. 2b is the reverse, where the PL dynamics without B-field (light blue) decay much faster than with B-field (dark blue), i.e., thus a decreasing rate with B-field. This contradiction between the observation and interpretation cast doubt on the whole manuscript.

We thank the reviewer for this potential confusion. We now discuss in more detail why we identify the *slow rate component of the biexponential decay* (see next comment) as the relevant one for the magnetic brightening. For this, we added a paragraph to the manuscript and a figure to the SI stating that the weight of the slow decay strongly increases (Page 9 manuscript, Fig. S5c, d):

“In contrast, the dark exciton emission shows a strong magnetic field dependence. We find the dynamics at zero field well-described by a biexponential decay with a dominant component with time constant $\tau_{\text{short}} \sim 1$ ns (close to $\tau_{\text{IRF}} \sim 0.7$ ns) and a low-weight component with $\tau_{\text{long}} \sim 14$ ns, characteristic for dark exciton emission. We separately integrate the two exponential decays over time and find that the intensity of the fast component remains constant with magnetic field, while the intensity of the slow component increases approximately sixfold (Fig. S5c, d). As this increase agrees well with the observed intensity increase in steady state PL measurements, we confirm the slow decay as the dark exciton emission. Under applied magnetic field, the PL emission rate increases from 0.07 ns^{-1} to 0.11 ns^{-1} with approximately quadratic dependence on field strength (Fig. 2b inset), while the weight of the decay increases \sim tenfold (Fig. S5d inset). Due to its higher weight, the slow component dominates the signal at high fields, so that the total PL dynamics decay slower, even though the emission rate of the slow decay increases.”

We note that the trend becomes more obvious in Mn:PEPI where the further enhanced PL rate leads to a faster overall decay with B field (Fig. 2d).

Figure S5: Transient circularly polarized PL of PEAPbI films at 4 K. **c**, dark exciton biexponential fit and its two components. **d**, time integrated PL for the fast and slow component as a function of magnetic field. While the fast component intensity remains unaffected by the magnetic field, the slow component increases \sim sixfold (due to higher weight A_{slow} , inset), which identifies the slow component as the dark exciton emission.

2. Moreover, If the Authors use a bi-exponential fitting in explaining the dynamics, I suggest the Authors either report each lifetime component and give interpretation to them or report the rate as the effective lifetime. However, this should be clearly stated in the manuscript together with the rationale of why the Authors do so. The Authors should also show that such analysis would not affect the further interpretations down the line.

This question connects to Question 1 from the Reviewer. As discussed in detail above, we added a paragraph and a supporting figure to explain why the long-lived PL emission is the relevant one for our analysis (Page 9 manuscript, Fig. S5c-d):

1. At zero field, a long-lived emission with little weight fits the characteristics of the dark exciton
2. The \sim 6-fold increase in time-integrated trPL signal of the slow decay component matches the magnetic brightening observed in steady-state PL measurements. In contrast, the fast component

time-integrated trPL signal stays constant with magnetic field, so cannot explain the observed brightening.

Regarding the interpretation of the fast component (of lifetime ~ 1 ns), we note that the time constant is close to that of the instrument response function (0.7 ns) and emphasize that the physical origin of the fast component is of no importance for the analysis of our main findings, since we clearly identify the slow component as the relevant one for our report of magnetic brightening of dark excitons.

3. Given the comments #1 and #2 can be answered, I like the Authors' new interpretation of two possible mechanisms. Could I suggest the Authors also do quantitative modelling on their interpretations to fit their experimental data and provide the quantitative values for the spin-dependent XD formation rate and/or XD recombination rate, based on the 2 proposed mechanisms?

We agree with the reviewer that quantitative modelling will help to confirm one of the proposed mechanisms and to improve our understanding of the system. However, this would require experimental data on ultrafast exciton dynamics within the first nanosecond, which is currently beyond the scope of our setups, which renders any modelling highly speculative. We therefore consider it more appropriate to present the modelling in following work, in which we plan to investigate the spin dependent relaxation mechanism using more specialized experimental setups currently under construction.

Unless the Authors could provide a satisfactory answer on comments #1 and #2, I could not recommend this paper for publication. On a side note, if the case if this work could be published in the future, I believe the data presentation could be further improved (e.g., Fig. 3a is too cluttered). Moreover, I believe some figures in the SI should also go to the main text (e.g., Fig. S3 and Fig. S6a-b).

We thank the reviewer again for their constructive comments #1 and #2, which we now answered and which helped us improving our manuscript, both in presentation and content. While we agree that Fig. S3 and S6 support our findings, but since both show magneto-PL data (already presented in the manuscript in Fig. 2) we prefer to leave them in the SI to avoid overloading the manuscript. We have further improved the readability of Fig. 3a, as suggested.

Reviewer #2 (Remarks to the Author):

I appreciate the authors for performing additional experiments and adding more data. The additional data certainly helped identifying the origin of the emission peaks in the spectra. Although many points from the earlier review are clarified, I am concerned about the possible mechanisms of CPL of dark exciton the authors presented in this revision.

We thank the reviewer for appreciating our efforts in taking additional experimental data that helped strengthen the claims of our manuscript.

The PL assigned to dark exciton in PEPI (Fig. 3a) shows no CPL in contrast to bright exciton, which seems consistent with singlet dark state ($J=0$) and triplet bright state ($J=1$) reported by Efros and coworkers (Nature volume 553, pages189–193(2018)) for cesium lead halide perovskites. In contrast, the possible mechanisms for CPL in Mn:PEPI the authors presented, and in Fig.4, dark exciton with different spin states are mentioned implying nonzero J for dark exciton in Mn:PEPI. Although the detailed spin-dependent physics may be done in future, the feasibility of dark exciton to have different spin states should be discussed, because it is crucial for understanding the mechanism of the polarization control which is the central theme of this study. Of course, 2D layered perovskite may have different exciton fine structure from 3D nanocrystals studied in the work by Efros and coworkers. If this structural difference results in nonzero J for dark exciton, it should be discussed, although the data from PEPI seems to suggest J is still zero in 2D perovskite.

We thank the reviewer for their comment regarding the total angular momentum J of the dark exciton state. We agree that the detailed spin-dependent physics are better left for future work, but we thank the reviewer for the suggestion that a clarification of the dark exciton spin states is helpful for the reader. We extended the discussion of the state in the manuscript (Pages 7, 13):

“Recent studies on exfoliated PEPI single crystals reported that both bright and dark exciton are each split into two substates with ~ 1 meV energy difference by strong electron-hole exchange interaction, thus yielding an exciton fine structure of four optically-active states.⁴⁵ However, since this splitting is likely even smaller in our less confined bulk material and due to larger PL linewidth, we are not able to resolve this fine structure.”

“Although the polarization of the dark exciton emission in Mn:PEPI follows the magnetization of the material, no giant Zeeman splitting of the PL peaks is observed for the doped and undoped sample. Only a

small Zeeman shift is resolved under high magnetic fields (> 20 T, Fig. S4a), confirming the existence of spin substates for the dark exciton and suggesting the possibility of a non-zero angular momentum of this state, also in agreement with previous reports on the exciton fine structure in PEPI.”

Figure S4: PL energy and polarization at 4 K. a, PL peak energy for both circular polarizations of Mn:PEPI: X_D with a ~ 100 ms pulsed magnetic field in Faraday geometry and excitation at 405 nm. A small Zeeman shift of the dark exciton emission only becomes measurable for $B > 20$ T.

While the singlet/triplet nature of excitons has been established in single nanocrystals, we agree with the reviewer that the exciton fine structure is strongly dependent on the structure and chemical composition of the material. For example, in the cited work from Efros, a triplet ground state was identified in CsPbBr_3 nanocrystals, while Lounis et al. (Nature Materials 18, 717 (2019)) experimentally found a singlet ground state in FAPbBr_3 nanocrystals. We therefore agree that a definite statement about the nature of exciton states in PEPI might be not conclusive, as the reviewer emphasizes. To the best of our knowledge, the most comprehensive study on exciton fine structure in PEPI is that by Xiong et al. (Nano Lett. 20, 7, 5141–5148 (2020)) where they study exfoliated single crystals. Although their material (exfoliated single crystal vs drop-casted film in our case) and some of their observations differ from ours, their reported exciton fine structure is compatible with our proposed mechanism: They assign their experimentally observed two peaks to two excitons X_2 (our X_D) and X_3 (our X_B), of which each is split into a lower and upper state due to the strong (confinement-enhanced) electron – hole exchange interaction, yielding X_2^L , X_2^U , X_3^L , X_3^U where the splitting between X_2^L and X_2^U is ~ 1 meV. Considering the stronger dielectric confinement in exfoliated layers, it is reasonable to assume that this splitting is even smaller in our material and thus not possible to resolve in our experiments. To confirm the existence of the two spin substates in our system, we performed magneto-PL up to 66 T and indeed see a small Zeeman splitting at very high fields (which we have now added as Fig. S4a). We are confident that the

added reference and detailed discussion of the work by Xiong et al., together with the presented high-field measurements added in the revised manuscript, gives sufficient evidence for the likely existence of spin sub-states of the dark exciton in our materials. We thank the reviewer once more for helping us to improve our manuscript here.

Reviewers' Comments:

Reviewer #1:

Remarks to the Author:

The Authors have answered all my concerns on the physics.

On the side note, I still believe that the figures quality on the work can be further improved to present the manuscript better.

One example would be the y-axis label in Fig. 2a and c. It would be better to put "sigma+ PL emission", rather than just "PL emission". That way it will be clearer to the reader, rather than have to infer it from the captions.

Fig. 2a and c's insets, and in Fig. 3a, a horizontal line at $y=0$ could be added as eye-guide.

Use +14T and -14T, instead of 14T and -14T, etc.

Reviewer #2:

Remarks to the Author:

I am satisfied with the revision.

Reply to Reviewer Comments

Reviewer #1 (Remarks to the Author):

The Authors have answered all my concerns on the physics.

On the side note, I still believe that the figures quality on the work can be further improved to present the manuscript better.

One example would be the y-axis label in Fig. 2a and c. It would be better to put " σ^+ PL emission", rather than just "PL emission". That way it will be clearer to the reader, rather than have to infer it from the captions.

Fig. 2a and c's insets, and in Fig. 3a, a horizontal line at $y=0$ could be added as eye-guide.

Use +14T and -14T, instead of 14T and -14T, etc.

We thank the reviewer for the suggestions regarding the figure quality and implemented them accordingly.

Reviewer #2 (Remarks to the Author):

I am satisfied with the revision.

We thank the reviewer for the positive assessment.